# Relative uptake of carbonyl sulphide to CO2: insights from a coupled boundary layer - canopy inverse modelling framework

Peter J. M. Bosman<sup>1</sup>, Maarten C. Krol<sup>1,2</sup>, Laurens N. Ganzeveld<sup>1</sup>, Felix M. Spielmann<sup>3</sup>, and Georg Wohlfahrt<sup>3</sup>

Correspondence: Peter J. M. Bosman (peter.bosman.publicaddress@gmail.com)

Abstract. Carbonyl sulphide (COS) is an atmospheric trace gas that has been suggested as a proxy to estimate carbon uptake by plants. To this end, the concept of leaf relative uptake (LRU), the ratio of deposition velocities of COS and CO<sub>2</sub>, has been introduced to obtain plant CO<sub>2</sub> uptake fluxes from COS flux measurements. In our study we use a coupled soil – canopy – atmospheric mixed layer model to simulate CO<sub>2</sub> and COS uptake by vegetation explicitly, and derive LRU. In this modelling framework, the exchange of COS is coupled to the exchange of H<sub>2</sub>O and CO<sub>2</sub> via stomatal conductance. The latter is calculated using an A-gs (Assimilation-stomatal conductance) photosynthesis model, accounting for separate exchange at sunlit and shaded leaves. Despite limited complexity, our coupled model include most key processes involved in daytime land atmosphere exchange. The models are embedded in an inverse modelling framework, allowing for a structured model parameter estimation. We performed a parameter optimisation for a boreal forest in Finland (Hyytiälä), using observation data from July 2015. We took a holistic approach and aimed to obtain model parameters consistent with a large set of observations, including COS and CO<sub>2</sub> molar fractions (measured in and above the canopy) and fluxes. By optimising parameters, we obtained a good fit to many observation types simultaneously. Analysing the corresponding modelled LRU, we found strong within-canopy variations at the leaf scale, with highest LRU values for shaded leaves near the bottom of the canopy. These variations can be explained to a large extent by differences in photosynthetically active radiation (PAR), vapour pressure and leaf temperature. Based on these findings, we propose a new parameterisation of canopy-scale LRU based on absorbed PAR and vapour-pressure deficit of sunlit leaves near the canopy top. We performed several additional optimisations, without re-optimising leaf exchange parameters: two for the same location, but for the months August and September, and two for a needleleaf forest in Austria (Mieming). We obtained a generally good fit with observations in all of these optimations, suggesting transferability of model parameters to different months and locations. When testing the LRU parameterisation using Hyytiälä model data from August and September (data not used for deriving the parameterisation), the results of the physical model were well-approximated, although observations suggest somewhat lower LRU values for a large part of the day. For Mieming, the parameterisation also provided a satisfactory fit to the physical model. For both locations we found that the LRU of sunlit leaves near the top of the canopy provides a good approximation of the canopy-scale LRU. Our results provide insight in the behaviour of LRU in the canopy, and the new parameterisation, based on both absorbed PAR and VPD, can contribute to improving COS-based ecosystem plant carbon uptake estimates in needleleaf ecosystems, but further validation is needed.

<sup>&</sup>lt;sup>1</sup>Meteorology and Air Quality Group, Wageningen University, Wageningen, the Netherlands

<sup>&</sup>lt;sup>2</sup>Institute for Marine and Atmospheric Research, Utrecht University, Utrecht, the Netherlands

<sup>&</sup>lt;sup>3</sup>Department of Ecology, University of Innsbruck, Innsbruck, Austria

45

50

#### 1 Introduction

The uptake of carbon dioxide by forests and other land vegetation plays a key role in regulating the climate on earth. It is therefore beneficial to have a good knowledge of these large fluxes. Ecosystem-scale photosynthetic  $CO_2$ -fluxes are traditionally estimated from eddy covariance (EC) measurements, e.g. mounted on a measurement tower at some height above the forest canopy. However, this is not a direct measurement of canopy photosynthesis. The EC-system measures the net  $CO_2$ -flux, which includes not only vegetation uptake but also respiration coming from the underlying soil surface, as well as from above-ground plant organs. Further processing of EC-measurements is needed to separate the net  $CO_2$  flux into gross primary productivity (GPP) and ecosystem respiration with this approach (Reichstein et al., 2005), introducing uncertainty. An alternative approach is the use of measurements of carbonyl sulphide (COS) fluxes, an atmospheric trace gas that is taken up by plants through their stomata. The advantage is that there is usually no large concurrent emission flux of COS near the location of the uptake flux (Whelan et al., 2018). The main uptake of COS in higher plants is due to hydrolysis after entering the leaves, catalysed by the enzyme carbonic anhydrase (Protoschill-Krebs et al., 1996). This leads to the production of  $H_2S$  and  $CO_2$  (Ferm, 1957). The canopy net photosynthesis flux ( $F_{CO_2,veg}$ , mol COS m<sup>-2</sup> s<sup>-1</sup>) can be derived from a known ecosystem-scale vegetation net COS uptake flux ( $F_{COS,veg}$ , mol COS m<sup>-2</sup> s<sup>-1</sup>) using the following formula:

$$F_{\text{CO}_2,\text{veg}} = \frac{F_{\text{COS},\text{veg}}}{\text{LRU}_{\text{can}}} \frac{[\text{CO}_2]}{[\text{COS}]}$$
(1)

Herein,  $[CO_2]$  and [COS] are measured (molar) concentrations  $[mol\,m^{-3}]$  of  $CO_2$  and COS respectively (or mole fractions in  $[mol\,mol^{-1}]$ ) and  $LRU_{can}$  (leaf relative uptake at canopy scale) is the ratio of COS and  $CO_2$  deposition velocities. Note that  $F_{CO_2,veg}$  is the net canopy photosynthesis, which is not identical to GPP (see discussion in Wohlfahrt et al. (2012)).

An important source of uncertainty in this approach arises from uncertainty in the value of LRU<sub>can</sub> and its spatial and temporal variability. For instance, it is known that light and humidity have an effect on the leaf relative uptake on the leaf scale (Kooijmans et al., 2019). This can be expected to lead to significant in-canopy variability of the relative uptake (Sun et al., 2022). As LRU<sub>can</sub> integrates the uptake of the whole canopy, variability in LRU<sub>can</sub> due to differences in environmental variables can be expected as well. This has also been reported in a field study at an agricultural field (Maseyk et al., 2014). Plant experiments in a glasshouse indicated species-specific effects of drought on the relative uptake of COS and CO<sub>2</sub> (Spielmann et al., 2025). A further difficulty is the presence of soil COS fluxes (Sun et al., 2015). In case these fluxes are significant, measured ecosystem-scale COS fluxes ( $F_{COS}$ ) require a correction to obtain  $F_{COS,veg}$ . Despite these difficulties, the LRU concept has been used to provide ecosystem estimates of photosynthesis fluxes (Asaf et al., 2013), including a recent estimate of global terrestrial GPP (Lai et al., 2024). Blonquist et al. (2011) also used LRU<sub>can</sub> to derive GPP, but without using direct COS flux measurements. They instead used LRU<sub>can</sub> with mole fractions and mole fraction gradients of COS and CO<sub>2</sub> to scale the net CO<sub>2</sub>-flux to GPP. Another application of COS is the estimation of stomatal conductance on the canopy scale. Wehr et al. (2017) found a good agreement between conductances estimated using COS and conductances estimated using another independent method.

We focus in this integrative study on (Scots pine-dominated) needleleaf forests, as we have two extensive datasets at our disposal, and given the large area covered by needleleaf forests on a global scale. We employ a coupled model, consisting of an atmospheric boundary layer model, a plant canopy model and a soil model. The resulting coupled model describes the exchange of H<sub>2</sub>O, CO<sub>2</sub> and COS between the lower atmosphere and the underlying ecosystem. The coupled model is overall relatively simple, but still includes most key processes involved in land atmosphere exchange over forest areas during daytime. The model is embedded in an inverse modelling framework, allowing for a structured parameter optimisation using observations. We use diverse observations (temperature, fluxes, humidity, mole fractions at multiple heights) to optimise parameters related to the (coupled) atmosphere–biosphere exchange of H<sub>2</sub>O, CO<sub>2</sub>, and COS. Thereby we aim to obtain a model parameter set that is consistent with a diverse set of observation streams, and is applicable to needleleaf forests in general.

Using our optimised parameter set, we model the relative uptake of COS and  $CO_2$  within a boreal forest canopy, thereby accounting for influences of environmental variables on leaf fluxes. Making use of the model (output), we analyse the environmental drivers of within-canopy relative uptake variability. Based on this we propose a parameterisation for  $LRU_{can}$ , that uses variables that are relatively easy to estimate. Our aim is to have a parameterisation that is applicable for needleleaf forests in general. Lai et al. (2024) (referred to as Lai24) used a parameterisation for  $LRU_{can}$ , derived in Kooijmans et al. (2019), that is based on measurements of the leaf-scale relative uptake of COS and  $CO_2$  at a boreal forest location, to estimate global terrestrial GPP. Their estimate, 157 ( $\pm 8.5$ ) PgC yr<sup>-1</sup>, is significantly higher than most remote sensing estimates (e.g. Beer et al., 2010; Jung et al., 2020; Lai et al., 2024). The Lai24 parameterisation is based solely on photosynthetically active radiation (PAR), while more environmental variables seem to influence LRU (Kooijmans et al., 2019; Sun et al., 2022). We will investigate how well the Lai24 parameterisation performs on the canopy scale, and to what extent the Lai24 parameterisation is transferable to another needleleaf forest. We also provide a different parameterisation specifically for the canopy-scale relative uptake. Thereby we aim to contribute to improving COS-based GPP estimates for needleleaf forest regions.

We try to answer the following main research questions in this paper:

- 1. Can we obtain a set of model parameters that is applicable to Scots pine-dominated forests, or perhaps even needleleaf forests in general?
  - 2. How does the relative uptake of COS and CO<sub>2</sub> vary within the canopy, and what drives the variability?
  - 3. Can, using our framework, a parameterisation for LRU<sub>can</sub> be constructed that performs better than the Lai24 leaf-scale-based parameterisation?

In Sect. 2 we present our methods and the data we use, including the (inverse) modelling framework (Sect. 2.1) and specifically the canopy model (Sect. 2.2). In Sect. 3 we present the results of the optimisation of model parameters using observations from a boreal forest (Hyytiälä, Finland). We analyse how the relative uptake of COS and CO<sub>2</sub> varies throughout the Hyytiälä canopy in the model (Sect. 3.1.4). Later we describe a new parameterisation for the leaf relative uptake at the canopy scale, obtained from our model with optimised parameters (Sect. 3.2). In Sect. 3.3 we apply the framework to a needleleaf forest in Austria, to test the generality of the (photosynthesis and leaf COS uptake) model parameters and LRU<sub>can</sub> parameterisation obtained with Hyytiälä data. In Sect. 4 we provide a discussion on the results.

## 2 Methods and data

# 2.1 Inverse modelling framework

The framework we use in this paper is called ICLASS-can, which is an extension of the ICLASS framework (extensively described in Bosman and Krol, 2023). The ICLASS framework can be used to study the exchange of gases, moisture, heat, and momentum between the land surface and the lower atmosphere. The general aim of the framework is to allow the assimilation of various streams of observations (fluxes, mole fractions at multiple heights, etc.) to estimate model parameters, thereby obtaining a model that is consistent with a diverse set of observations (Bosman and Krol, 2023). In an optimisation, a cost function is minimised. This cost function usually contains two parts. One part contains the difference between model output and observations, the second (optional) part contains the difference between the parameter values and the prior estimates of these parameters. ICLASS is a variational Inverse modelling framework for the (slightly adapted) Chemistry Land-surface Atmosphere Soil Slab model (CLASS, Vilà-Guerau De Arellano et al., 2015). We have added a relatively simple canopy model to the ICLASS framework, in order to simulate gases and atmospheric conditions in forest canopies in more detail. The resulting coupled forward model consists of an atmospheric mixed layer model, coupled to a new canopy model, which is in turn coupled with a simple soil model. For temperature and moisture, the soil model distinguishes between upper soil and deeper soil. It has a dynamic upper soil temperature and moisture content, used for calculating soil respiration. For COS, the soil is treated in more detail, using a soil diffusion-reaction model for COS (Sun et al., 2015). The atmospheric mixed layer model assumes that turbulence is vigorous enough to result in a well-mixed layer. Therefore, there is just one value for scalars such as mixed-layer potential temperature and CO<sub>2</sub> mole fraction. Because of this assumption, we do not model nighttime conditions in this study. The mixed-layer height is dynamic. There is exchange taking place between the mixed layer and both the underlying canopy and the free troposphere above. In our configuration, above-canopy surface layer scalars are calculated employing Monin-Obukhov similarity theory (Monin and Obukhov, 1954; Stull, 1988). The coupled model has no horizontal dimension, but a constant mixed-layer advection can be prescribed. The mixed-layer model provides the best results on days with prototypical mixed layer behaviour, i.e. days on which advection is either absent or uniform in time and space, deep convection and precipitation are absent, and sufficient incoming shortwave radiation heats the surface (Bosman and Krol, 2023). We provide a brief description of the new canopy model in the next section, and a more detailed description can be found in the Supplement.

# 2.2 Canopy model SiLCan

SiLCan stands for Simplified Layered Canopy. The model simulates four different tracers in the canopy, namely CO<sub>2</sub>, COS, moisture and temperature. The canopy is layered, and the user defines the number of layers. Exchange between the layers is parameterised in a simple way, with eddy-diffusivity exchange coefficients. For photosynthesis, the A-gs-approach (Jacobs, 1994; Ronda et al., 2001) at leaf scale is followed, thereby explicitly simulating leaf scale CO<sub>2</sub> fluxes, separately for sunlit and shaded leaves. Leaf or plant area densities are used to scale up fluxes from the leaf scale to the canopy layer scale, depending on the considered flux. The stomatal conductances for CO<sub>2</sub> are calculated by A-gs, and are linearly related to

Figure 1. Sketch of the canopy model used in this manuscript.

those for COS and H<sub>2</sub>O (different diffusivities). Stomatal conductances thus link the leaf fluxes of CO<sub>2</sub>, COS and H<sub>2</sub>O. Leaf boundary layer conductances are calculated for the three before-mentioned gases, taking differences in diffusivity into account. For COS, an internal resistance is calculated (following Cho et al., 2023). Incoming shortwave radiation at the top of the canopy is calculated by CLASS, SiLCan uses this to calculate (absorbed) PAR in each layer, separately for sunlit and shaded leaves. Outgoing longwave radiation from a leaf surface is calculated based on the Stefan–Boltzmann law. In our configuration, absorbed incoming longwave radiation is calculated as the incoming longwave radiation at the top of the canopy (calculated by CLASS), multiplied with a constant leaf emissivity and a factor s<sub>LWin</sub> (constant in space and time) that we optimise. The energy balance is calculated at leaf level, leading to a leaf (skin) temperature which is used in the calculation of the sensible heat fluxes and H<sub>2</sub>O fluxes of leaves (sunlit and shaded separately). In our configuration we set heat storage in the leaves to zero, the energy balance is calculated using only modelled radiation and sensible and latent heat flux terms. A sketch of the canopy model is shown in Fig. 1, and elaborate details can be found in the Supplement.

# 2.3 Optimisations using Hyytiälä data

140

145

Hyytiälä (Fluxnet ID FI-Hyy) is a forest location in Finland, at 61.85 °N, 24.28 °E, and 181 m above sea level. The forest is a Scots pine stand (Pinus sylvestris) sown in 1962 (Launiainen et al., 2011), with some other tree species present as well (Vesala et al., 2022), (supplementary material of Kooijmans et al., 2019). A measurement tower is present at the location. More details on the location can be found in Launiainen et al. (2007, 2011). In all simulations, we use a time step of 60 seconds, and divide the canopy in 17 layers. The vertical canopy structure is represented by layers with a depth of  $\approx$ 1 m except for the top and bottom canopy layers that have a depth of  $\approx$ 1.5 m to properly resolve the exchange with the soil and atmospheric mixed layer (avoiding potential numerical problems with large fluxes in small layers). The soil COS model (Sun et al., 2015) was run on a smaller time step of 10 seconds, to prevent numerical instability. Advection of all scalars was set to zero in all simulations.

We aimed to minimise the mismatch between model output and observations, albeit with the constraint of taking prior information about the parameters into account (minimising a cost function, containing an observation and prior information part). In a first optimisation, we used observations from July 2015, in total 26 different observation streams (CO<sub>2</sub> mole fractions

165

at different heights, above-canopy sensible and latent heat flux, ...; full list in Table A1). We selected the daytime window within 4 and 16 UTC (7 and 19 h local time). Part of the observation streams we used are shown in Figure 2. The observations represent averages over an hour, and we further averaged these observations over multiple days, to obtain a representative average and reduce the influence of processes such as time-varying advection. For this averaging, we selected the 8 days that have the highest mean PAR between 4 and 16 UTC. Measurement errors were estimated as the standard deviation of the observations over the 8-days we average (e.g. the measurement error of the CO<sub>2</sub> mole fraction at 125 m at 10.30 UTC is the standard deviation of the 8 CO<sub>2</sub> 125-m mole fraction values at 10.30 UTC). For some measurement errors we used less than 8 data points, as there is some missing or insufficient quality data. Observational errors are constructed from these measurement errors and specified model errors (Bosman and Krol, 2023).

The 25 parameters we optimised (the state) are listed in Table 1, together with prior and posterior values and the prior standard deviations. These parameters include e.g. free-tropospheric lapse rates of potential temperature and humidity, the initial soil water content of the top soil layer, and a constant that is important for soil respiration. We also included some parameters of the A-gs photosynthesis model and a parameter scaling the internal conductance of COS ( $\alpha_{giCOS}$ ). Thus, the state has an effect on the leaf relative uptake of COS and CO<sub>2</sub>. To limit the complexity of the optimisation problem, we did not optimise all parameters of the A-gs model. The prior parameter values were chosen as reasonable guesses. The prior variances for the A-gs parameters were chosen large (Table 1), allowing sufficient freedom for the optimisation algorithm, and limiting the influence of the subjective prior guesses.

In subsequent optimisations we used data from August and September 2015, averaged in a similar way as the July 2015 data. For those optimisations we used the posterior parameters from the July optimisation as prior parameter guesses. We optimised the same variables as for July, apart from  $A_{\text{m,max,ref,toc}}$ ,  $a_{\text{d}}$ ,  $K_{\text{b}}$ ,  $g_{\text{m,ref}}$ ,  $f_{0}$ ,  $\varepsilon_{0}$ ,  $\alpha_{\text{giCOS}}$ ,  $\alpha_{\text{wind scale}}$  and  $V_{\text{SU,max}}$  (description in Table 1). For these variables we stick to the optimised values from July, as we aim to find values for these parameters that are transferable to other months and locations.  $R_{10}$  was kept in the state, as the observations suggest a stronger respiration flux in (the averaged periods in) August and September compared to July.

The shape of the leaf area density profile for all simulations was first estimated based on Fig. 1 of Launiainen et al. (2011). As all-sided leaf area index (LAI) in Hyytiälä is roughly between 6.5 and 7 in the period Jul-Sep 2015 based on Fig. 1 of Vesala et al. (2022), the leaf area densities were then scaled by a factor, identical for all layers, to make total LAI (summed over the model layers) equal to 6.5.

For analysing posterior correlations of the parameters we optimised, we performed an ensemble of parameter optimisations, consisting of 127 members. For each member, the prior parameters and the model-observation differences are perturbed. The prior information part of the cost function was disabled in the ensemble. Details of the correlation analysis procedure can be found in Bosman and Krol (2023).

## 2.4 Optimisations using Mieming data

Mieming is a forest location in Austria (Fluxnet ID AT-Mmg, 47°18.9938′N, 10°58.2053′E) at 960 m above sea level. A 30-m measurement mast is present at the location. Close to the site a mountain range is present, the station is located on a gently

sloped plateau (Platter et al., 2024). This potentially complicates the thermodynamics and flow. Scots pine is the dominant tree species, with substantial Juniperus (Juniperus communis) in the understorey. More details on the site can be found in Platter et al. (2024).

The leaf area density profile was estimated from a lidar scan of the site, and scaled such that the all-sided leaf area index becomes 4 (measurements indicate  $3.4 \pm 0.6$ ). We use the same time resolution and a similar spatial resolution as for Hyytiälä, now dividing the (somewhat lower) canopy into 12 layers. We use averaged data for August 2023 in a first optimisation, and for July 2023 in a second optimisation. Data were again averaged over days with high mean PAR (the 8 days in the month with highest mean PAR between 7 and 19 local time, LT), such that we get hourly observations for one daytime period, from 7.30 to 18.30 LT. Measurement errors are estimated by the same approach as for Hyytiälä. As Scots pine dominates both Hyytiälä and Mieming, we use the A-gs parameters (and  $\alpha_{\rm giCOS}$ ) obtained from the optimisation in Hyytiälä for simulating Mieming. Besides these parameters, the set of parameters we optimise (Table 2) is to a large extent similar to those for Hyytiälä (Table 1). The posterior parameters from the Hyytiälä optimisation for July 2015 serve as prior parameter estimates for the Mieming optimisations.

On some of the days with high PAR that were selected for averaging observations, we also have branch bag measurements, containing leaf COS and CO<sub>2</sub> fluxes and mole fractions, for sunlit leaves in the upper canopy. These measurements are not assimilated in the optimisation, but we use these for comparing modelled leaf-scale relative uptake of COS and CO<sub>2</sub> (Sect. 2.5) with observations.

# 2.5 Relative uptake COS and CO<sub>2</sub>

The leaf relative uptake at canopy scale (LRU<sub>can</sub>), as introduced in Eq. 1, cannot directly be derived from eddy covariance flux observations, as COS can also be taken up by the soil, the measured CO<sub>2</sub> flux includes respiration and storage fluxes can be present. The ecosystem relative uptake, defined below, can however easily be derived from eddy covariance flux observations and observed concentrations (or mole fractions):

$$ERU = \frac{F_{COS,eco}}{F_{CO_2,eco}} \frac{[CO_2]}{[COS]}$$
 (2)

wherein  $F_{\text{COS},\text{eco}}$  is the flux above the canopy, measured by eddy covariance. We use here eddy covariance COS and CO<sub>2</sub> fluxes that are not storage corrected, given that the modelled COS flux to which we compare the  $F_{\text{COS},\text{eco}}$  observations, is the instantaneous flux between the top canopy layer and the mixed layer. To calculate ERU in Hyytiälä, we use mole fractions at 125 m height, both in the observations and the model. This height is chosen since we have observations of both CO<sub>2</sub> and COS mole fractions at this height. The mole fractions at the levels of the leaves might however be different from those at 125 m height. For calculating ERU in Mieming, we use mole fractions at 20 m height.

For Hyytiälä, we can derive LRU<sub>can</sub> from the observations. To this end, we subtract the measured soil COS flux from the eddy covariance CO<sub>2</sub> flux. We assume respiration and COS emission from above-ground sources to be small, and neglect it in the calculation. Next to that, we correct the eddy covariance observations for storage below the sensor (in contrast to the calculation for ERU), as we are interested in plant fluxes. Note

that for well-mixed daytime conditions the storage fluxes are thought to have a small influence (Kohonen et al., 2020). The formula for LRU<sub>can</sub> is given in Eq. 1. The LRU<sub>can</sub> error bars are estimated from the spread in LRU<sub>can</sub> values over the days we use data from. As an example, the error bar for LRU<sub>can</sub> at 10.30 UTC indicates the standard deviation of the LRU<sub>can</sub> values at 10.30 UTC for the different days we average the observations over (8 days, or less in case of missing or bad quality data). For Mieming, we do not have all required measurements to derive LRU<sub>can</sub>. For calculating modelled LRU<sub>can</sub> in Mieming we use modelled COS and CO<sub>2</sub> mole fractions at 20 m height.

We define the leaf-scale relative uptake as:

$$LRU_{leaf} = \frac{F_{COS,leaf}}{F_{CO_2,leaf}} \frac{[CO_2]}{[COS]}$$
(3)

wherein  $F_{\text{COS,leaf}}$  [mol m<sup>-2</sup> s<sup>-1</sup>] is the leaf-scale flux of COS and [COS] is the concentration of COS [mol m<sup>-3</sup>] (or mole fraction) outside the leaf boundary layer. Similarly,  $F_{\text{CO}_2,\text{leaf}}$  is the leaf-scale flux of CO<sub>2</sub> and [CO<sub>2</sub>] is the concentration of CO<sub>2</sub>. All four components are within the same canopy model layer (or for the same branch in case of observations). The calculation is performed in all model layers, and separately for sunlit and shaded leaves. For a theoretical analysis of the leaf-scale relative uptake, see Wohlfahrt et al. (2012). For our analysis of in-canopy LRU<sub>leaf</sub> differences in Sect. 3.1.4, we further expand the equation above. Using the model equations (for our configuration) given in the Supplement, Eq. 3 can be written as:

$$LRU_{leaf} = \frac{r_{tot,CO2}}{r_{tot,COS}} \frac{[CO_{2}]}{[CO_{2}] - [CO_{2}]_{int}} = \frac{r_{b,CO2} + \frac{1}{\frac{1}{r_{s,CO2}} + \frac{1}{r_{cut,CO2}} \frac{1}{and CoS}}}{\frac{1.56}{1.37} r_{b,CO2} + \frac{1}{\frac{1}{1.21} r_{s,CO2} + r_{int,COS}} + \frac{1}{r_{cut,CO2} \frac{1}{and CoS}}} \frac{1}{1 - \frac{[CO_{2}]_{int}}{[CO_{2}]}} \approx \frac{r_{b,CO2} + r_{s,CO2}}{\frac{1.56}{1.37} r_{b,CO2} + r_{int,COS}} \frac{1}{1 - \frac{[CO_{2}]_{int}}{[CO_{2}]}}$$
(4)

wherein  $r_{\text{cut}}$  is the cuticular resistance,  $r_{\text{b}}$  is the leaf boundary layer resistance,  $r_{\text{int,COS}}$  is the internal resistance for COS,  $r_{\text{s}}$  is the stomatal resistance,  $r_{\text{tot}}$  is the total resistance and  $[\text{CO}_2]_{\text{int}}$  is the internal CO<sub>2</sub> concentration. The approximation in the last part of the equation concerns the assumption that the cuticular pathways for CO<sub>2</sub> and COS uptake are negligible (e.g. Berry et al., 2013). When additionally neglecting the leaf boundary layer resistance, the equation above becomes equal to equation (8) of Seibt et al. (2010).

What is clear from the equation above is that any environmental variable that influences stomatal resistances, the COS internal resistance, the boundary layer resistance, or the internal CO<sub>2</sub> concentration can influence LRU<sub>leaf</sub> (e.g. PAR, VPD, wind speed,...). We simulate the canopy profiles of environmental variables. We thereby take effects of canopy structure (leaf and plant area density distribution) on environmental variables into account.

The model variables that vary between sunlit and shaded leaves are the leaf temperature and the amount of absorbed PAR. To find out the relative importance of both factors in determining LRU differences between sunlit and shaded leaves, we performed the following model experiment: we took a sunlit leaf in the top layer, and prescribe for this leaf the leaf temperature of a shaded leaf in the same layer, without changing the absorbed PAR of the leaf. We subsequently investigated how the modelled LRU of the leaf changed. Next, we took again a sunlit leaf in the top model layer, and prescribed it the absorbed PAR of a shaded leaf, without changing the leaf temperature. Thus, we performed a univariate sensitivity analysis.

The elementary model variables that vary between leaves in the top and bottom layer, and have an influence on LRU, are the  $CO_2$  mole fraction, vapour pressure, air temperature, leaf temperature, amount of absorbed PAR,  $A_{m,max,ref}$  (a leaf photosynthesis parameter, see Table 1) and wind speed. Note that COS mole fraction is not included, as it cancels out in our LRU equation, given that in the model COS uptake at the leaf scale is a linear function of the COS mole fraction. We again performed a univariate sensitivity analysis, now replacing one-by-one the values of the 7 relevant variables from a shaded top-layer leaf in the model with those of a shaded leaf in the bottom layer.

## 3 Results




# 3.1 Optimisation Hyytiälä July 2015

We first take a look at the observations and the performance of the prior (i.e. before optimisation) and posterior (i.e. after optimisation) models. We than briefly analyse posterior state changes in the optimisation. To gain some insight in the relevant in-canopy physics, we analyse vertical canopy profiles of the posterior model simulation, before moving on to the relative uptake of COS and CO<sub>2</sub>.

# 3.1.1 Observations and model performance

Fig. 2 shows 10 of the 26 assimilated observation streams. The CO<sub>2</sub> mole fractions, both in the canopy (Fig. 2a) and above the canopy (Fig. 2c), generally decrease in the morning, in the observations and in the models. This is (in the models) caused partly by entrainment of air from above the mixed layer, and partly due to the effect of vegetation uptake. The COS mole fraction observations (Fig. 2b, d) show a less clear trend. The above-canopy flux of CO<sub>2</sub> (Fig. 2e) peaks around midday, both in the observations and the posterior model. The observed COS flux (Fig. 2f) is more noisy, but seems to peak earlier than the CO<sub>2</sub> flux. Both in the observations and the posterior model, the temperature just below the canopy top (Fig. 2h) increases during the day, and slightly drops at the end of the simulation period. The specific humidity just below the canopy top (Fig. 2g) is higher in the morning as in the afternoon, both in the posterior model and the observations. Temperature and humidity just below the canopy top are predicted well within the  $1-\sigma$  error bars. The posterior fit with observations is generally greatly improved compared to the prior (Fig. 2). The total cost function reduces from a value of about 1317 to about 77. The reduced chi-square goodness-of-fit statistic ( $\chi_r^2$ , Bosman and Krol, 2023), accounting for the number of observations and parameters, equals 0.39. This indicates the model fits the observations well. Besides quantifying the fit of the total optimisation, we can also consider a specific observation stream using the partial reduced chi-square value ( $\chi_{r,j}^2$  for the jth observation stream, Bosman and Krol, 2023). The results for each observation stream individually are shown in Table A1. The values are generally low, indicating the posterior model fits most observation streams well. The observation stream that has the best posterior fit in terms of the above quantity is the temperature at 67 m height. The observation stream that has the worst posterior fit in terms of the above quantity is the specific humidity at the bottom of the canopy (Table A1). Both the modelled latent (Fig. 2j) and sensible heat flux (Fig. 2i) are somewhat on the low side. Overall, the coupled model reproduces the averaged July 2015 data well.







# 3.1.2 Adjustments to the state

The full state that we optimised is shown in Table 1. The soil COS uptake capacity  $(V_{\rm SU,max})$  is strongly increased, as the prior model underestimated the soil COS flux (not shown). The scaling factor for the internal conductance of COS  $(\alpha_{\rm giCOS})$  is increased by about 25%. As shown in Fig. 2e and j, the prior model has too strong above-canopy  ${\rm CO_2}$  and latent heat fluxes, compared to observations. These fluxes are reduced in the posterior model, and are sensitive to stomatal conductance. Therefore it is no surprise that the calculated stomatal conductances are generally reduced compared to the prior simulation (e.g. for sunlit top leaves the decrease in stomatal conductance is 76% on average). Several parameters from the state influence the calculated stomatal conductances, and the parameter adjustments are not always trivial to interpret. For instance, in the posterior state,  $\varepsilon_0$  (maximum initial quantum use efficiency) is strongly reduced, while  $A_{\rm m,max,ref,toc}$  (related to photosynthetic capacity, Table 1) is increased by more than 50%. These changes have contrasting effects on the stomatal conductances, although the strength of their effects in time might differ. As we expect some of the (photosynthesis) parameters to be correlated, we computed the posterior correlations of the parameters we optimised (based on an ensemble of optimisations, Fig. A1, Sect. 2.3). We indeed see a negative correlation between  $A_{\rm m,max,ref,toc}$  and  $\varepsilon_0$ , though not very strong (-0.37).

We find the strongest correlations (-0.92) to be between  $\gamma_{COS}$  (free-tropospheric COS lapse rate) and  $\Delta_{COS}$  (initial COS jump between mixed layer and free troposphere), and between  $h_{init}$  (initial mixed-layer height) and  $\Delta_{CO2}$ . Thus, optimisations with a large posterior value of  $\gamma_{COS}$  tend to have a low posterior value of  $\Delta_{COS}$  and vice-versa. This likely indicates that similar results can be obtained by increasing  $\gamma_{COS}$  as by increasing  $\Delta_{COS}$ . From a physical point of view this makes sense, as an increase in any of both parameters tends to entrain more COS from the free troposphere into the mixed layer, although their effects in time might differ. Similarly, these parameters can potentially compensate for advection, which is set to zero in all our simulations.

As a result of the correlations present, it will be difficult to confidently determine (some of) the individual parameters, and a combined subset of parameters is likely more robust. In the optimisations for August and September (Sect. 2.3) we will keep the optimised photosynthesis parameters and  $\alpha_{giCOS}$  (Table 1), as we aim to find values of these parameters (parameter subset) that can be transferred to other months and locations.

#### 3.1.3 Vertical model profiles canopy

In this section we analyse vertical canopy profiles of the posterior model simulation. The fraction of sunlit leaves strongly decreases towards the bottom of the canopy (Fig. 3c), as a result of interception of light by the plants. When inspecting the modelled vertical profiles of net leaf-level photosynthesis (Fig. 3b) around noon, we observe that the strongest CO<sub>2</sub> uptake takes place at the top of the canopy, and sunlit leaves have a much stronger uptake than shaded leaves. This can to a large extent be explained by the profiles of absorbed PAR (Fig. 3b). The stomatal conductance for sunlit leaves shows a local minimum near the middle of the canopy (Fig. 3a). Clearly, this does not follow the profile of absorbed PAR for sunlit leaves, with lowest absorbed PAR values at the bottom of the canopy, and highest values in the top canopy. However, the minimum in stomatal conductance near the middle canopy practically coincides with a maximum in vapour pressure deficit (VPD, Fig. 3a) for sunlit





leaves near the middle canopy. Modelled stomatal conductance is sensitive to VPD. The minimum in stomatal conductance near 7 m height can be interpreted as a (modelled) response of plants to the high VPD, to prevent too much water loss.

Why does VPD for sunlit leaves maximise in the middle canopy? The mole fraction of water vapour (Fig. 4b) is highest at the bottom of the canopy, providing a reason why VPD decreases in the bottom compared to the middle of the canopy. This does however not explain why VPD is higher in the middle compared to the top canopy. The higher VPD in the middle canopy is explained by the vertical profile of leaf temperature (Fig. 4b): The modelled leaf temperature for sunlit leaves is highest in the bottom canopy. At the same time, modelled air temperature shows only small variability compared to sunlit leaf temperature. Somewhat counter-intuitively, the high leaf temperatures in the middle to bottom of the canopy coincide with the lowest values of available leaf net radiation and leaf sensible heat flux (for sunlit leaves, Fig. 4c). The reason for this apparent contradiction can be found in the leaf boundary layer resistance (Fig. 4d). The way-higher boundary layer resistance in the middle of the canopy compared to the top means that a larger leaf-to-air temperature gradient is needed for the same leaf sensible heat flux. The leaf boundary layer resistance profile only depends on wind speed (decreasing resistance with higher wind speed, see canopy model description in Supplement), which decreases sharply from the top to the middle model canopy (Fig. 4d). Thus, even though boundary layer resistances are much smaller than stomatal resistances, the vertical profile of wind speed still has a strong influence on the modelled vertical stomatal conductance profiles for sunlit leaves, via VPD. It can be noted that modelled leaf temperature for sunlit leaves exceeds air temperature by multiple degrees in Fig. 4 (b and c), up to almost 5 °C. Similar temperature differences have been measured between needles and air (Martin et al., 1999, and references therein). This suggests that the modelled leaf boundary layer resistances in Fig. 4d are in a plausible order of magnitude.

The shape of the stomatal conductance profiles resembles the shape of the (absolute value of the) COS leaf flux profiles, with strongest uptake at the top and a local minimum in uptake in the middle canopy. Note that the shape of the profiles differs from that of CO<sub>2</sub>. A major reason is that for CO<sub>2</sub> the gradient between internal and air concentration plays an important role. The difference between internal and ambient CO<sub>2</sub> peaks (for sunlit leaves) in the middle canopy (not shown), close to the location of the local minimum in stomatal conductance, partially compensating the effect of the low stomatal conductance. For COS, the internal concentration is assumed zero. The internal resistance for COS follows the profiles of leaf temperature, with consequently more or less opposite shapes for sunlit and shaded leaves (Fig. 4d). However, the internal resistance for COS is, both for shaded and sunlit leaves, smaller than the stomatal resistance, limiting the influence of internal resistance differences on the COS flux profiles.

The leaf-scale water vapour fluxes are highest at the top of the canopy (Fig. 4a). The main drivers for this flux are VPD and stomatal conductance. When scaling up vegetation fluxes from leaf to layer or canopy scale, the leaf area density (Fig. 3c) becomes important. As an example, the location of the peak in the layer-total vegetation  $H_2O$  flux does not correspond to the location of the peaks in the leaf-scale  $H_2O$  fluxes (Fig. 4a). Similarly, the location of the peaks in the leaf-scale sensible heat fluxes does not correspond to the location of the peak in the layer-total vegetation sensible heat flux (Fig. 4c). For calculating the latter flux, plant area density (including branches etc.) is used instead of only leaf area density.

Clearly, the plant fluxes of COS, CO<sub>2</sub> and H<sub>2</sub>O inside the canopy are relatively complex and are driven by the interplay of many variables. It is important to realise that in our model the exchange of these gases is fully coupled. As we gained insight in






what happens inside the (model) canopy in terms of photosynthesis and COS uptake, we now shift our attention to the relative uptake of COS and CO<sub>2</sub>.

# 3.1.4 LRU<sub>leaf</sub> inside the Hyytiälä canopy

To better understand what drives variability in  $LRU_{can}$ , the variable that is commonly used to estimate canopy net photosynthesis (Eq. 1), we will first analyse the relative uptake within the canopy at the leaf scale, using results (at 11 h LT) of the simulation of the optimised model for Hyytiälä for July 2015. From the model output containing fluxes and mole fractions at all layers within the canopy, we calculate the leaf-scale relative uptake. Inspecting the derived  $LRU_{leaf}$  (Eq. 3) at different times of day (Fig. 5), we observe a strong variation within the canopy and between sunlit and shaded leaves. The shaded leaves have a notably higher  $LRU_{leaf}$ , and  $LRU_{leaf}$  is higher at the bottom of the canopy compared to the top. To increase our understanding of the  $LRU_{leaf}$ , we now analyse the reasons for these differences in the model.

We first focus on the difference between sunlit and shaded leaves in the top model layer of the canopy. The results of the first sensitivity analysis (model experiment described in Sect. 2.5) indicate that the amount of absorbed PAR is by far dominant in determining LRU differences between sunlit and shaded leaves (Fig. 6a). The direct effect (in the model) of the low absorbed PAR of the shaded leaves is a reduction in stomatal conductance, i.e. an increase in stomatal resistance. The effect of the increased stomatal resistance differs between COS and CO<sub>2</sub>. The impact for COS is smaller as for COS there is also an internal resistance. Consequently, in Eq. 4, the numerator is almost directly proportional with stomatal resistance, while the relative increase of the denominator is less, due to the significant internal resistance of COS (although smaller than stomatal resistance, see also Fig. 4d and Fig. 3a). This leads to an increase of LRU<sub>leaf</sub> for shaded leaves.

A second difference visible in Fig. 5 is the higher LRU<sub>leaf</sub> for leaves near the bottom of the canopy, in particular for shaded leaves. To find out the relative importance of the various environmental factors in shaping this difference, we performed a similar model experiment as before (Sect. 2.5, second sensitivity analysis), but now involving top and bottom layer differences. The results of this experiment (Fig. 6b) indicate that changes in vapour pressure are the most important, followed by the amount of absorbed PAR. Leaf temperature differences are relevant as well. In contrast to absorbed PAR, differences in vapour pressure and leaf temperature directly affect multiple variables in Eq. 4, namely internal CO<sub>2</sub> concentration, internal conductance for COS and stomatal conductance.

The above analysis indicates that amount of absorbed PAR, leaf temperature and vapour pressure are important for governing LRU at the leaf scale. In Sect. 3.2, we propose a parameterisation for LRU<sub>can</sub> linked to these variables.

# 3.1.5 Canopy relative uptake for Hyytiälä

When analysing canopy relative uptake (LRU<sub>can</sub>, Eq. 1, the quantity most relevant for estimating canopy net photosynthesis), the posterior model fit to observations is substantially better than the prior fit (Fig. 7a red full line vs yellow dashed line, posterior bias 0.27, RMSE 0.58). Most data points are fitted by the posterior model within one standard deviation. This is also the case for ERU (Eq. 2, not shown). In general, there is a small positive bias in LRU<sub>can</sub>, although the observational spread is large. Note here that LRU is a derived quantity that is not used as observation stream in the optimisation.


For comparison, we have also plotted  $LRU_{leaf}$  in Fig. 7a, for a (posterior) modelled sunlit top layer leaf and for a shaded bottom leaf. These represent the lowest and highest  $LRU_{leaf}$  values in the canopy respectively (Fig. 5). The modelled  $LRU_{can}$  corresponds more to the sunlit top layer leaf than to the shaded bottom leaf.  $LRU_{can}$  is approximated well when weighting  $LRU_{leaf}$  throughout the canopy with the  $CO_2$  uptake flux, while a weighting with sunlit and shaded leaf area index in all canopy layers (differing LAI values per layer) gives too high values. This can be explained by the fact that a much larger proportion of modelled  $CO_2$  and COS uptake takes place in the sunlit upper vs the shaded lower canopy (Fig. 8).

# 3.2 Predicting LRU<sub>can</sub> for needleleaf forests

Our aim is to obtain a parameterisation for LRU<sub>can</sub> that is applicable to needleleaf forests in general, and is based on independent variables that are relatively easy to estimate. Using also the information from Sect. 3.1.4, we select canopy-integrated amount of absorbed PAR (PAR<sub>abs</sub>) and vapour pressure deficit of sunlit top (upper canopy model layer) leaves (VPD<sub>sun,top</sub>) as independent variables for our regression model. These variables can be estimated or approximated based on remote sensing products (e.g. Myneni et al., 2002; Olofsson and Eklundh, 2007; Nolan et al., 2016) or global re-analysis products (e.g. Fang et al., 2022). Note that in the within-canopy analysis above, leaf temperature was also identified as an important variable. However, leaf temperature is used in the calculation of VPD, and we can expect leaf temperature to correlate in time with absorbed PAR. For simplicity, we chose a linear regression model. As training data we use the model output from the optimised model for July 2015 in Hyytiälä. We obtain the following linear regression equation:

$$LRU_{can} = 2.52 - 1.07 \times 10^{-3} PAR_{abs} - 0.399 VPD_{sun,top}$$
(5)

wherein  $PAR_{abs}$  has the units  $Wm_{\rm ground}^{-2}$  and  $VPD_{sun,top}$  is in kPa. The equation above has an  $R^2$  of 0.98 on the training data. The two mentioned variables thus capture most of the variability in  $LRU_{can}$ . Note that we use  $PAR_{abs}$  and  $VPD_{sun,top}$  from the model output as input variables for our regression model.

To test to what extent the regression equation holds outside the training data set, we re-optimised our model using observations from August 2015. We re-optimised variables relating to the time-specific meteorological situation of the month, but did not re-optimise photosynthesis or leaf COS uptake parameters (we keep those parameters as the July-optimised values, see Sect. 2.3).

In this new optimisation, we obtain again a satisfactory fit to many observation streams, with a cost function that reduces from about 565 to 125 ( $\chi^2 = 0.66$ ). The performance of our regression model trained on the July data applied to the August data, is shown in Fig. 7b. Note that we use PAR<sub>abs</sub> and VPD<sub>sun,top</sub> from the posterior model output as input variables for our regression model, as we do throughout this manuscript. Comparing the linear regression LRU<sub>can</sub> model with the physical model, the R<sup>2</sup> is 0.96 and the root mean squared error is 0.04. The regression model thus provides a very good approximation to the results of the physical model. Comparing with the (LRU<sub>can</sub> derived from) observations, the R<sup>2</sup> is 0.23 and the root mean squared error is 0.85. There is generally a positive bias, although limited when taking the spread in observations (error bars) into account. In Fig 7, we also show the results of the leaf-scale-based parameterisation from Kooijmans et al. (2019), used in Lai et al. (2024) (referred to as Lai24). This parameterisation, to which we provide observed PAR at 18 m height as only input

variable, clearly performs well also on the canopy scale for this data. Comparing with the observational  $LRU_{can}$ , the  $R^2$  is 0.48 and the root mean squared error is 0.70.

To test how well the regression equation performs in somewhat more different conditions, we performed a new optimisation for September 2015. Climatologically, September has a lower temperature and less incoming shortwave radiation compared to July. We optimised the same variables as for the August optimisation. In this optimisation, we obtain an acceptable fit to many observation streams, with a cost function that reduces from about 594 to 148 ( $\chi^2 = 0.78$ ). The CO<sub>2</sub> and COS fluxes are generally slightly underestimated by the model, although the fit is for most data points within the 1- $\sigma$  error bars.

The performance of the regression model for September is shown in Fig. 7c. Comparing the linear regression model with the physical model, the  $R^2$  is 0.76 and the root mean squared error is 0.15, indicating a good fit. Comparing with the observations, the  $R^2$  is -0.20 and the root mean squared error is 0.57. The regression model from Lai et al. (2024) performs very well in September as well. Comparing with the observational LRU<sub>can</sub>, the  $R^2$  is -6.52 and the root mean squared error is 1.42 (strongly influenced by the first data point in the morning).

The regression model for LRU<sub>can</sub> thus approximates the physical model well, also for months outside the training data. Comparing with (LRU<sub>can</sub> derived from) observations, differences are larger, although still limited when taking the spread in observations into account. In the next section we will analyse the results of applying the framework and parameterisation to a different location.

# 3.3 Optimisations Mieming





We now apply our inverse modelling framework to a needleleaf forest at a more southerly location (Mieming, Austria) to check how universal the optimised photosynthesis parameters and LRU<sub>can</sub> parameterisation are. We therefore use the leaf exchange parameters we obtained for July 2015 in Hyytiälä, and do not include these parameters in the state we optimise (Table 2). We first discuss the optimisation using data from August 2023.

In general, we can fit the Mieming observations well (Fig. 9, showing the 10 assimilated observation streams for the July optimisation). As for Hyytiälä, the posterior fit with observations is improved compared to the prior. The cost function reduces from a value of about 45 to about 25 ( $\chi^2 = 0.20$ ). The relative reduction in cost function is a lot smaller as for the July 2015 Hyytiälä optimisation (prior 1377, posterior 77). The prior model already predicts the above-canopy  $CO_2$ , COS and  $H_2O$  fluxes relatively well (Fig. 9e, f, j), suggesting reasonable transferability of the Hyytiälä leaf exchange parameters. This optimisation is less constrained as those for Hyytiälä, given the lack of e.g. soil flux observations. The optimised parameters are shown in Table 2. Since we do not have soil fluxes, we also do not have observations of  $LRU_{can}$ , albeit we can derive the ERU from the observations (Fig. A2). The fit between our modelled ERU (based on optimised parameters) and observed ERU is acceptable, given the huge spread in observations. We also applied the parameterisation for  $LRU_{can}$ , obtained from the July Hyytiälä optimisation, to Mieming. We find a satisfactory agreement between the linear regression model (Fig. 10) and the physical model. Except for the early morning, the difference between the physical and linear regression models is always smaller than 0.3. For the optimisation using data from July, this is also the case (Fig. 10b). For both months, the parameterisation from Lai24 (to which we now provide observed PAR at 30 m height as only input variable) underestimates the physical model

LRU<sub>can</sub>, and performs less well than our regression model. As for Hyytiälä, the difference between modelled LRU<sub>can</sub> and the modelled LRU<sub>leaf</sub> of a sunlit top leaf is rather small. In Fig. 10 we also compare the modelled LRU<sub>leaf</sub> for a sunlit top leaf with observations of LRU<sub>leaf</sub> for (mostly) sunlit leaves in the middle to top crown layer of the canopy. For July, there is generally an underestimation by the model, although not as strong as the underestimation by the Lai24 parameterisation. For August, the fit is relatively good for 10 to 13 h, in the afternoon the underestimation is relatively large. There was however only one day of measurements available for August, which might not be fully representative for the modelled period.

We performed an additional optimisation for Mieming to test the generality of the leaf exchange parameters obtained with Hyytiälä data. In this optimisation we included the set of leaf exchange parameters that was included in the July 2015 Hyytiälä optimisation (see Table 1) in the state that we optimise. We find relatively strong adjustments in some of those parameters, the strongest relative changes occur for  $\varepsilon_0$  (maximum initial quantum use efficiency, + 42%) and  $a_{\rm d}$  (description in Table 1, + 33%). However, the optimisation results in a  $\chi^2$  value that is only slightly lower than the one for the posterior model with Hyytiälä parameters (0.17 vs 0.20), described earlier. Re-optimising therefore seems to provide limited benefit for simulating the observations well. The prediction of LRU<sub>can</sub> only slightly changes in this optimisation (Fig. A3), with largest changes in the early morning.

# 4 Discussion


# 460 **4.1 Model performance**

For Hyytiälä, we have optimised a set of parameters making use of not less than 26 different observation streams. Because of this, we obtain model parameters that are consistent with a large number of observations, instead of over-focusing on one observation stream. This more holistic approach of modelling achieves a good fit with observations, despite the simplifications present in the model. Major simplifications are the exchange between canopy layers and the exchange between the canopy and the overlying mixed layer. The model uses exchange coefficients, thereby assuming that the local gradients drive the turbulent exchange. This neglects the influence of larger scale turbulent eddies (see e.g. Brunet, 2020; LeMone et al., 2019). Sweep and ejection events are important for canopy flow and exchange (see e.g. Zhu et al., 2007), e.g. CO<sub>2</sub> and H<sub>2</sub>O can be ejected from the understory into the atmosphere above by such events (Moonen et al., 2025). We tried to smooth out the influence of these processes (acting on short timescales), by using averaged observations.

The above-mentioned simplifications are a likely reason why the specific humidity was not fitted well (Table A1) near the bottom of the canopy. The change in  $LRU_{leaf}$  due to vertical within-canopy vapour pressure differences (Fig. 6) might consequently be overestimated. However, the exact values of the changes in  $LRU_{leaf}$  are not relevant for our analysis that served to identify variables relevant for inclusion in the  $LRU_{can}$  parameterisation.

A close inspection of the A-gs model equations (Supplement Sect. S1.8.1) reveals that the internal CO<sub>2</sub> concentration ([CO<sub>2</sub>]<sub>int</sub>) does not directly respond to photosynthesis and thus PAR. From a physiological point of view, [CO<sub>2</sub>]<sub>int</sub> is linked to PAR, as PAR supplies the energy and reduction equivalents for the carboxylation process, and thus influences the 'photosynthetic demand' for CO<sub>2</sub>, which reflects in the internal CO<sub>2</sub> concentration. Under sufficient amounts of radiation, this should




be of limited importance (Ronda et al., 2001, and references therein). Under low radiation conditions, this might be more important, and thus it would be interesting for future studies to investigate how the modelled LRU would change when using a different photosynthesis model that represents the joint effects of diffusional supply of and photosynthetic demand for  $CO_2$  on  $[CO_2]_{int}$ .

The measurement tower in Mieming is located on a gently sloping plateau with mountains nearby, complicating the thermodynamics and flow situation. We left out the observations of the  $CO_2$  mole fraction at 2 m height, as they showed a very different time evolution to those at 20 m height. Such contrasting evolutions are very difficult to reproduce with our 1d model, when using realistic parameter values. Despite the simplification of the in-canopy physical processes in our model, the COS and  $CO_2$  mixing ratios at 20 m and above-canopy fluxes (important for  $LRU_{can}$ ) are reproduced relatively well (Fig. 9). We therefore believe that our relatively simple model adequately represents most key processes relevant for the combined daytime exchange of COS,  $CO_2$ , and  $CO_2$  between the canopy and the atmosphere.

# 4.2 LRU<sub>can</sub> and its parameterisation

The linear regression model for LRU<sub>can</sub> approximates the physical model very well, both for Hyytiälä and Mieming. The fit with Hyytiälä observations is acceptable, although there is a positive bias and the variation is somewhat underestimated. Note that we do not directly optimise LRU<sub>can</sub>, but we assimilate the ecosystem CO<sub>2</sub> and COS fluxes and molar fractions above canopy, which directly or indirectly link to LRU<sub>can</sub> (Eq. 1). Given the complexity of this variable (Eq. 1), some extent of mismatch between model and observations seems logical. As is clear from the error bars (Fig. 7), the differences in LRU<sub>can</sub> between the 8 days over which we average are substantial, complicating the comparison with physical and regression models. Analysing the error bars of the 4 LRU<sub>can</sub> related components in Fig. 2c-f for Hyytiäla, makes clear that the COS flux causes an important part of the variation.

Our new regression model is based on canopy absorbed PAR and VPD of top sunlit leaves. A theoretical analysis by Sun et al. (2022) also indicated a dependence of LRU on PAR and humidity. The shapes of our canopy profiles correspond well to the shape of the LRU profiles for a hypothetical canopy in their Fig. 8.

LRU<sub>can</sub> is a canopy integrated quantity, and we have seen (Sect. 3.1.4) that strong within-canopy variations occur. Theoretically, LRU<sub>can</sub> should be larger than the LRU<sub>leaf</sub> of sunlit top leaves, as also shaded leaves, which have a higher LRU<sub>leaf</sub> (Fig. 5), contribute to LRU<sub>can</sub>. The difference between LRU<sub>can</sub> and LRU<sub>leaf</sub> of sunlit top leaves should increase when the contribution of shaded leaves to the total COS and CO<sub>2</sub> exchange of the canopy increases. However, as is clear from our model results in Fig. 7 and Fig. 10, the LRU of sunlit leaves is not very different from the total LRU of the canopy, especially when omitting the (early) morning and evening. This is related to the limited contribution of shaded leaves to the COS and CO<sub>2</sub> exchange (Fig. 8 for Hyytiälä). The similarity between LRU<sub>can</sub> and LRU<sub>leaf</sub> of sunlit top leaves is encouraging for the use of canopy COS fluxes to estimate canopy CO<sub>2</sub> uptake, as it suggests that leaf-scale measurements of LRU could be used to estimate LRU<sub>can</sub>. This also likely explains why the parameterisation used in Lai et al. (2024) (derived in Kooijmans et al. (2019)) performs remarkably well for Hyytiälä, given that it was derived using (sunlit) leaf-scale observations from this site. For locations like Hyytiälä, leaf-scale measurements can be used for upscaling, provided these mirror what sunlit leaves are doing. It however remains to





be seen whether this also works for tall canopies with large LAI (and consequently likely a larger shaded leaf area fraction), e.g. tropical rain forests.

However, for Mieming the Lai24 parameterisation clearly underestimates the leaf-scale observations, and our parameterisation outperforms Lai24 at this location. One could hypothesise that the difference in performance between our and the Lai24 parameterisation could be (partly) related to the inclusion of VPD in our parameterisation. In Mieming the VPD can be expected to be higher, due to climate differences, including more intense solar radiation. In our posterior July optimisations, the modelled VPD of sunlit top leaves is about 40% higher in Mieming as in Hyytiälä at 14 LT. Compared to Hyytiälä, the modelled larger VPD in Mieming reduces LRU<sub>can</sub> as predicted by our linear regression model by about 0.29. Consequently, the higher VPD in Mieming cannot be the explanation for the lower LRU<sub>can</sub> for Mieming in Lai24 compared to our regression model, as the inclusion of VPD in our regression model reduces the difference between both parameterisations. Within the modelled period, the difference in magnitude of LRU<sub>can</sub> between Lai24 and our physical model is small in the early morning and evening, the difference is much larger later in the morning and during midday, when incoming PAR is higher. This suggests that the response to PAR in Lai24 is too strong for Mieming. It also has to be noted that for the July observations in Mieming, PAR reaches values above 2000  $\mu$ mol m<sup>-2</sup>s<sup>-1</sup> around noon. This is outside the range of values used in Kooijmans et al. (2019) to derive the Lai24 parameterisation at the Hyytiälä site (see their supplementary Fig S6).

Lai et al. (2024) applied the Lai24 parameterisation globally, obtaining a GPP of 157 ( $\pm 8.5$ ) PgC yr<sup>-1</sup>. However, for July and August in Mieming, the (derived) leaf-scale LRU<sub>leaf</sub> observations are on average about 75% higher than the Lai24 parameterisation results (Fig. 10). Applying a 75% higher LRU<sub>can</sub> to COS fluxes (Eq. 1) translates in a 75% smaller canopy net photosynthesis flux ( $\approx$  GPP) around midday in this location. Given that our model results (and theory, see Wohlfahrt et al., 2025) indicate that LRU<sub>can</sub> is still somewhat higher than the LRU<sub>leaf</sub> of sunlit top leaves (Fig. 7 and Fig. 10), the underestimation by Lai24 will be even larger. We therefore conjecture that the global GPP estimate provided by Lai et al. (2024) is very uncertain, as similar biases could be present in other ecosystems as well. This can be especially the case when vegetation structure, vegetation properties and environmental conditions are very different compared to Hyytiälä.

Our developed parameterisation, although not performing equally well as Lai24 in Hyytiälä, shows better transferability to Mieming. For the same reasons as mentioned above, one should be careful in applying our parameterisation to other locations, especially to other ecosystems than needleleaf forests. Our inverse modelling framework is well-suited to improve knowledge on the relative uptake of COS and CO<sub>2</sub> for different ecosystems. However, extensive measurement data need to be available, and currently these datasets are sparse.

# 540 4.3 Universality photosynthesis parameters

Using the optimised leaf exchange (A-gs and  $\alpha_{giCOS}$ ) parameters we obtained for July, we fitted the observations for August and September in Hyytiälä, and the observations in Mieming, to a satisfactory level. This suggests a level of general applicability of these parameters.

As described in Sect. 3.3, we performed an additional optimisation for Mieming in which we included (as state parameters) the set of leaf exchange parameters that was included in the July 2015 Hyytiälä optimisation. We found strong changes in the

parameters. However, given the correlations present between the A-gs parameters (derived from an ensemble of optimisations, Fig. A1, Sect. 2.3), it is however difficult to precisely determine the values of these parameters, it is likely that parameter equifinality causes multiple sets of parameters to perform comparably. Re-optimising seemed to provide limited benefit for simulating the observations well, as indicated by the small difference in  $\chi^2$  between the optimisations with and without the leaf exchange parameters in the state. This supports the applicability of the A-gs parameters obtained for Hyytiälä to Mieming, and the applicability is likely to hold for more needleleaf forest locations.

The problem of parameter equifinality could be tackled by reducing the complexity of the A-gs model, thereby reducing the parameter count. This would ideally lead to a more uniquely defined A-gs parameter set that is still transferable to multiple locations. Wohlfahrt et al. (2023) used an optimality model that requires fewer parameters than our A-gs model, which can be expected to lead to a more robust parameter set.

## 5 Conclusions





We have simulated daytime exchange of COS and CO2 over and within the canopy of needleleaf forest at two locations: Hyytiälä, Finland and Mieming, Austria. We used a coupled soil-atmospheric mixed layer-canopy inverse modelling framework, with the aim of optimising model parameters that govern biosphere-atmosphere exchange at the leaf scale. The results suggest a high level of transferability of leaf exchange model parameters between the two needleleaf forests. From the model results (7-19 LT) we found that the relative uptake of CO<sub>2</sub> and COS (LRU) at the leaf scale is highly variable within the canopy, with highest LRU values associated with shaded leaves at the bottom of the canopy. Our analysis indicates that the amount of absorbed photosynthetically active radiation (PAR), vapour pressure and leaf temperature are key variables determining this variability. As a next step, we developed a linear regression equation linking the model LRU at the canopy scale (LRU<sub>can</sub>) with canopy absorbed PAR and vapour pressure deficit at the top of the canopy. We found a good agreement between the regression and the physical model. For Hyytiälä, both the physical and regression model generally somewhat overestimated  $LRU_{can} \ with \ respect \ to \ the \ (noisy) \ observations. \ We \ found \ that \ the \ LRU \ of \ sunlit \ top \ leaves \ provides \ a \ relatively \ good \ estimate$ of LRUcan, which is encouraging for the use of canopy COS fluxes to estimate canopy CO2 uptake. At the same time, we find that the simple leaf-scale parameterisation obtained in Hyytiälä by Kooijmans et al. (2019), rolled out globally by Lai et al. (2024), does not perform well in a more southerly needleaf forest (Mieming, Austria). This suggests that climatic and canopy variability between different sites is relevant for LRU. We need more in-situ data across a wider range of species and climates to judge transferability of our developed LRU<sub>can</sub> parameterisation to different ecosystems and seasons. Our inverse modelling framework is well-suited to further improve knowledge on the relative uptake of COS and CO<sub>2</sub> for different ecosystems (e.g. scaling leaf to canopy or guiding data collection strategies). The results so far provide insights in the behaviour of LRU within the canopy, and are promising to improve canopy net photosynthesis estimates based on COS, especially for needleleaf forests.





Code and data availability. Much of the Hyytiälä data we used were obtained from the SmartSMEAR database that contains continuous data records from all SMEAR sites (https://smear.avaa.csc.fi/). The COS eddy covariance fluxes and mole fractions were provided by Linda Kooijmans and Kukka-Maaria Kohonen. Leaf-scale Hyytiälä data were obtained from https://zenodo.org/records/1211481#.XB4Lb9IzbIU. The (inverse) model code (the code of ICLASS-can) and the data and Python scripts we used for making figures and tables can be found at https://doi.org/10.5281/zenodo.17166145.

# Appendix A: Additional figures and table

Author contributions. MCK and PJMB mostly designed the study. PJMB performed the actual coding and (adjoint) model construction. PJMB performed the numerical optimisations and wrote the paper, the latter with extensive help from MCK. LNG assisted with interpreting results and provided extensive feedback during the study, and provided comments on the paper. FMS provided, prepared and discussed the Mieming data. GW provided feedback on the study and suggestions, and provided comments on the paper.

Competing interests. The contact author has declared that neither of the authors has any competing interests.

Acknowledgements. This work is part of the COS-OCS project (http://cos-ocs.eu/, last access: 04 August 2025), a project that received funding from the European Research Council (ERC) under the European Union's Horizon 2020 Research and Innovation programme (grant no. 742798). The numerical optimisations were carried out on the Dutch national e-infrastructure, with the support of SURF Cooperative. We thank Linda Kooijmans for providing us a lot of data, and for answering our questions about it. We also thank Kukka-Maaria Kohonen for providing us with some of the data. We also like to acknowledge the people responsible for the operation of the Hyytiälä and Mieming field stations, as the data from these stations was crucial to this research. We would also like to thank the people responsible for keeping the SmartSMEAR database operational. Also a big thank you to Magnus Bremer, for providing us a lidar-based plant area density profile estimate for Mieming. We hereby also thank Jordi Vilà-Guerau De Arellano (Wageningen University) and Arnold Moene (Wageningen University) for the interesting discussions. We also acknowledge Wu Sun (Carnegie Science) for the informative e-mail correspondence. We would also like to thank the developers of the CLASS model, and of the COSSM (soil COS) model.

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

**Table 1.** The prior and posterior parameter values in the Hyytiälä July 2015 optimisation, together with square root of prior variances.

| Parameter                                                       | Description                                     | Prior                  | Posterior              | $\sqrt{\text{Prior variance}}$ |
|-----------------------------------------------------------------|-------------------------------------------------|------------------------|------------------------|--------------------------------|
| $h_{ m init}$ [m]                                               | Initial mixed-layer height                      | 200.00                 | 517.31                 | 300.00                         |
| $\gamma_{\rm CO2}  [{ m ppm  m}^{-1}]$                          | Free-tropospheric CO <sub>2</sub> lapse rate    | $1.00 \times 10^{-2}$  | $8.42 \times 10^{-3}$  | $60.00 \times 10^{-3}$         |
| $\gamma_{\rm COS}  [{\rm ppb}{\rm m}^{-1}]$                     | Free-tropospheric COS lapse rate                | $3.00 \times 10^{-5}$  | $6.14\times10^{-6}$    | $1.00\times10^{-4}$            |
| $\gamma_q  [\mathrm{kg}  \mathrm{kg}^{-1}  \mathrm{m}^{-1}]$    | Free-tropospheric specific humidity lapse       | $-5.00 \times 10^{-6}$ | $-2.21\times10^{-6}$   | $3.00\times10^{-6}$            |
|                                                                 | rate                                            |                        |                        |                                |
| $\gamma_{\theta}  [\mathrm{K}  \mathrm{m}^{-1}]$                | Free-tropospheric potential temperature         | $6.00 \times 10^{-3}$  | $5.32\times10^{-3}$    | $5.00\times10^{-3}$            |
|                                                                 | lapse rate                                      |                        |                        |                                |
| $\gamma_{\mathrm{u}}  [\mathrm{s}^{-1}]$                        | Free-tropospheric zonal wind lapse rate         | $4.00 \times 10^{-3}$  | $-8.77 \times 10^{-4}$ | $4.00\times10^{-3}$            |
| $\Delta_{\rm CO2}$ [ppm]                                        | Initial CO <sub>2</sub> jump at mixed-layer top | -10.00                 | -10.98                 | 50.00                          |
| $\Delta_{\rm COS}$ [ppb]                                        | Initial COS jump at mixed-layer top             | $3.00 \times 10^{-2}$  | $3.22\times10^{-2}$    | $60.00 \times 10^{-3}$         |
| $\Delta_q  [\mathrm{kg}  \mathrm{kg}^{-1}]$                     | Initial specific humidity jump at mixed-        | $-1.50 \times 10^{-3}$ | $-2.37\times10^{-3}$   | $3.00\times10^{-3}$            |
|                                                                 | layer top                                       |                        |                        |                                |
| $\Delta_{\theta}$ [K]                                           | Initial pot. temp. jump at mixed-layer top      | 2.50                   | 1.18                   | 2.50                           |
| $\Delta_u [\mathrm{m  s}^{-1}]$                                 | Initial zonal wind jump at mixed-layer top      | 1.00                   | 1.84                   | 4.00                           |
| $lpha_{ m wind\ scale}\ [-]$                                    | Scaling factor wind extinction canopy           | 1.00                   | $5.91\times10^{-1}$    | 0.40                           |
| $R_{10} [\mathrm{mg_{CO2} \ m^{-2} s^{-1}}]$                    | Respiration at 10 °C and without water          | $8.00 \times 10^{-2}$  | $8.71\times10^{-2}$    | $2.00\times10^{-1}$            |
|                                                                 | stress                                          |                        |                        |                                |
| $w_{\rm g} \ [{ m m}^3 \ { m m}^{-3}]$                          | Volumetric water content top soil layer         | 0.35                   | 0.33                   | 0.10                           |
| $u  [\mathrm{m  s^{-1}}]$                                       | Initial mixed-layer zonal wind speed            | 4.50                   | 2.47                   | 3.50                           |
| $V_{\rm SU,max}  [{ m mol}  { m m}^{-3}  { m s}^{-1}]$          | Soil COS uptake capacity                        | 2.00                   | $1.90\times10^{1}$     | $1.00\times10^2$               |
| $K_{ m scale} [-]$                                              | Scaling factor for exchange coefficients        | 1.00                   | 1.93                   | 0.30                           |
| $s_{ m LWin} \left[ -  ight]$                                   | Multiplication factor incoming longwave         | 1.30                   | 1.21                   | 0.20                           |
|                                                                 | radiation vegetation vs top of canopy           |                        |                        |                                |
| $A_{\rm m,max,ref,toc} \left[{\rm mg_{CO2}m^{-2}s^{-1}}\right]$ | Top-of-canopy triose-phosphate-utilisation-     | 2.20                   | 3.76                   | 4.40                           |
|                                                                 | limited net rate of (leaf scale) photosynthe-   |                        |                        |                                |
|                                                                 | sis at 298 K (Van Diepen et al., 2022)          |                        |                        |                                |
| $K_{\rm b}  [{ m m}^2  { m m}^{-2}]$                            | Extinction coefficient for $A_{ m m,max,ref}$   | 0.29                   | 0.36                   | 0.25                           |
| $a_{\rm d}  [{\rm kPa}^{-1}]$                                   | Regression coefficient used to calculate the    | 0.07                   | 0.16                   | 0.14                           |
|                                                                 | value of vapour pressure deficit at which the   |                        |                        |                                |
|                                                                 | stomata close                                   |                        |                        |                                |
| $f_0$ $[-]$                                                     | See Supplement                                  | 0.89                   | 0.78                   | 1.78                           |
| $\varepsilon_0  [\mathrm{mg_{CO_2}  J^{-1}}]$                   | Maximum initial quantum use efficiency          | $1.70 \times 10^{-2}$  | $5.57\times10^{-3}$    | $3.40\times10^{-2}$            |
| $\alpha_{ m giCOS}$ [-]                                         | Parameter scaling internal conductance          | 1400.00                | 1763.31                | 2800.00                        |
|                                                                 | COS                                             |                        |                        |                                |
| $g_{ m m,ref}~[{ m mms}^{-1}]$                                  | Mesophyll conductance at 298 K                  | 7.00                   | 10.20                  | 14.00                          |

Figure 2. July 2015 Hyytiälä optimisation: prior (yellow dashed line) and posterior (full red line) model fit to the observations used in the cost function.  $\sigma_O$  and  $\sigma_I$  are the observational and measurement errors of the observations.  $CO2_{4.2}$  is the  $CO_2$  mole fraction at 4.2 m height above the surface,  $COS_{125}$  is the COS mole fraction at 125 m above the surface.  $F_{CO2}$  and  $F_{COS}$  are the  $CO_2$  and  $F_{COS}$  are the cost fluxes respectively above the top of the canopy, measured by an eddy covariance system.  $F_{COS}$  are the specific humidity and temperature at 16.8 m above the surface.  $F_{COS}$  are the above-canopy sensible and latent heat flux respectively.

Figure 3. Vertical profiles of variables related to photosynthesis. The profiles come from the optimised July 2015 model simulation. In (a) vapour pressure deficit (VPD) and stomatal conductance (gs) for CO<sub>2</sub> are shown. The stomatal conductances for COS or H<sub>2</sub>O can be obtained by multiplication with  $\frac{1}{1.21}$  or 1.6 respectively. In (b), we show leaf level net photosynthesis (An) and absorbed PAR (PARabs). Note that absorbed PAR is shown per square meter of all-sided (sunlit or shaded) leaf area. In (c) we show leaf area density (lad), plant area density (pad) and the fraction of sunlit leaf area. In contrast to plant area density, leaf area density includes only green leaf area, i.e. branches, dead leaves and stems are not included. 'sun' indicates sunlit leaf area and 'sha' indicates shaded leaf area. The model canopy height is 17 m, values are plotted at the location of the model node in each layer. Thus, the total LAI is not equal to the area to the left of the lad curve.

Figure 4. Vertical in-canopy (model and observation) profiles of various variables. In (a) we show the sunlit and shaded leaf fluxes of COS ( $F_{COS}$ ) and  $H_2O$  ( $F_{H2O}$ ), i.e. the leaf-air exchange per square meter all-sided leaf area, of COS and  $H_2O$  respectively. The total vegetation  $H_2O$  flux (tot  $F_{H2O}$  [W  $m_{ground}^{-2}$ ]) for each canopy layer is also plotted. In (b) the molar ratio of  $H_2O$  in the canopy is shown, together with sunlit and shaded leaf (skin) temperatures ( $T_s$ , we assume no difference between leaf and leaf skin temperature, as we do not account for heat storage in the leaves). (c) shows air temperature in the canopy, as well as the sensible heat flux for sunlit (H sun) and shaded leaves (H sha), and the vertical profile of net radiation at a sunlit leaf surface (H sun). The total vegetation sensible heat flux (H tot) for each canopy layer is also plotted, note that this flux has the units [W  $m_{ground}^{-2}$ ] in contrast to [W  $m_{all-sided \, leaf \, area}^{-2}$ ] for the other two heat fluxes. In (d) the boundary layer resistance for H0, shown (H1), as well as the horizontal wind speed inside the canopy (H1) and the internal resistance for H1. The modelled boundary layer resistances for heat can be obtained from those of H2. By multiplication with H3, are averages between 12–14 h LT. The modelled boundary layer resistances for heat can be obtained from those of H2.

**Figure 5.** Leaf relative uptake inside the canopy for the Hyytiälä July 2015 optimised model simulation. 'sun' and 'sha' indicated sunlit and shaded leaves respectively.

Figure 6. Results of two model experiments at 11 h LT: (a) shows the results of an experiment to determine contributions to differences in LRU between sunlit and shaded leaves inside the canopy (see text). We consider a sunlit leaf in the top layer. The differences ( $\Delta$ ) in the variables on the x-axis between sunlit and shaded top layer leaves (shaded — sunlit) are annotated in the figure. (b) shows the results of an experiment to determine contributions to differences in LRU between shaded top and bottom layer leaves inside the canopy (see text). We consider a shaded leaf in the top layer. The values of the differences in the variables on the x-axis between top and bottom layer shaded leaves (bottom — top) are annotated in the figure. The variables from left to right are:  $CO_2$  mole fraction, vapour pressure, air temperature, leaf temperature, absorbed PAR, wind speed,  $A_{m,max,ref}$  (relating to leaf photosynthesis, see Table 1) and a combination of all.

**Figure 7.** Modelling LRU<sub>can</sub> and predicting it by linear regression: (a) displays the results of the prior and posterior (physical) model for July 2015. LRU<sub>leaf</sub> for a sunlit top layer leaf and a shaded bottom layer leaf are also shown. The results of weighting LRU<sub>leaf</sub> with shaded and sunlit leaf area index in each canopy layer is also shown, as well as the result of weighting LRU<sub>leaf</sub> with the shaded and sunlit vegetation CO<sub>2</sub> uptake fluxes in each layer. In (b), the physical and linear regression model results for August 2015 are shown, together with the prediction from the leaf-scale regression equation used in Lai et al. (2024). The observations are shown by black stars. In (c) the results for September 2015 are shown. The first 10 timesteps (minutes) of the physical and new linear regression model are not shown, to reduce potential numerical noise. Note that the observation at 14.30h in (a) has no error bar, as there was only one valid data value at this time of day over the 8 days we averaged.

Figure 8. Relative cumulative vegetation COS and  $CO_2$  fluxes throughout the canopy, starting at 0 at the top of the canopy. The relative cumulative flux is defined as the fraction of the total (COS or  $CO_2$ ) vegetation flux over the whole canopy (sunlit + shaded). The cumulative shaded and sunlit fluxes sum to 1 at the bottom of the canopy, for COS and for  $CO_2$ .

**Table 2.** The prior and posterior parameter values in the August 2023 optimisation for Mieming, together with square root of prior variances.

| Parameter                                                                   | Description                                  | Prior                  | Posterior              | $\sqrt{\text{Prior variance}}$ |
|-----------------------------------------------------------------------------|----------------------------------------------|------------------------|------------------------|--------------------------------|
| h <sub>init</sub> [m]                                                       | Initial mixed-layer height                   | 517.31                 | 527.54                 | 300.00                         |
| $\gamma_{\rm CO2}  [{\rm ppm}{\rm m}^{-1}]$                                 | Free-tropospheric CO <sub>2</sub> lapse rate | $8.42 \times 10^{-3}$  | $-1.50 \times 10^{-2}$ | $60.00 \times 10^{-3}$         |
| $\gamma_{\rm COS}  [{\rm ppb}{\rm m}^{-1}]$                                 | Free-tropospheric COS lapse rate             | $6.14 \times 10^{-6}$  | $3.93\times10^{-5}$    | $1.00\times10^{-4}$            |
| $\gamma_q  [\mathrm{kg}  \mathrm{kg}^{-1}  \mathrm{m}^{-1}]$                | Free-tropospheric specific humidity lapse    | $-2.21 \times 10^{-6}$ | $-3.53 \times 10^{-6}$ | $3.00\times10^{-6}$            |
|                                                                             | rate                                         |                        |                        |                                |
| $\gamma_{\theta}  [\mathrm{K}  \mathrm{m}^{-1}]$                            | Free-tropospheric potential temperature      | $5.32 \times 10^{-3}$  | $6.36\times10^{-3}$    | $5.00\times10^{-3}$            |
|                                                                             | lapse rate                                   |                        |                        |                                |
| $\Delta_{\text{CO2}}$ [ppm]                                                 | Initial CO2 jump at mixed-layer top          | -10.98                 | -15.59                 | 50.00                          |
| $\Delta_{\rm COS}$ [ppb]                                                    | Initial COS jump at mixed-layer top          | $3.22 \times 10^{-2}$  | $5.88\times10^{-2}$    | $60.00 \times 10^{-3}$         |
| $\Delta_q  [\mathrm{kg}  \mathrm{kg}^{-1}]$                                 | Initial specific humidity jump at mixed-     | $-2.37 \times 10^{-3}$ | $-1.87 \times 10^{-3}$ | $3.00\times10^{-3}$            |
|                                                                             | layer top                                    |                        |                        |                                |
| $\Delta_{\theta}$ [K]                                                       | Initial pot. temp. jump at mixed-layer top   | 1.18                   | 4.69                   | 2.50                           |
| $R_{10} \left[ \text{mg}_{\text{CO}2} \text{ m}^{-2} \text{s}^{-1} \right]$ | Respiration at 10 °C and without water       | $8.71 \times 10^{-2}$  | $7.85 \times 10^{-2}$  | $2.00\times10^{-1}$            |
|                                                                             | stress                                       |                        |                        |                                |
| $u  [\mathrm{m  s^{-1}}]$                                                   | Initial mixed-layer zonal wind speed         | 2.47                   | 0.49                   | 3.90                           |
| $V_{\rm SU,max} \; [{ m mol}{ m m}^{-3}{ m s}^{-1}]$                        | Soil COS uptake capacity                     | $1.90 \times 10^{1}$   | $1.90\times10^{1}$     | $1.00\times10^2$               |
| $K_{ m scale} [-]$                                                          | Scaling factor for exchange coefficients     | 1.93                   | 1.94                   | 0.30                           |
| $s_{\mathrm{LWin}} [-]$                                                     | Multiplication factor incoming longwave      | 1.21                   | 1.20                   | 0.20                           |
|                                                                             | radiation vegetation vs top of canopy        |                        |                        |                                |
| [COS][ppb]                                                                  | Initial mixed-layer COS mole fraction        | 0.45                   | 0.47                   | 0.10                           |

**Figure 9.** Model fit to the measurements from the Mieming location for August 2023. The yellow line is the prior model, the red line is the posterior model.  $\sigma_O$  and  $\sigma_I$  are the observational and measurement errors of the observations.  $CO2_{20}$  is the  $CO_2$  mole fraction at 20 m height above the surface,  $COS_{20}$  is the COS mole fraction at 20 m above the surface.  $F_{CO2}$  and  $F_{COS}$  are the  $CO_2$  and  $F_{COS}$  are the  $F_{COS}$  are the  $F_{COS}$  and  $F_{COS}$  and  $F_{COS}$  are the  $F_{COS}$  and  $F_{COS}$  and  $F_{COS}$  are the  $F_{COS}$  and  $F_{COS}$  are the  $F_{COS}$  and  $F_{COS}$  and  $F_{COS}$  are the  $F_{COS}$  are the  $F_{COS}$  are the  $F_{COS}$  and  $F_{COS}$  are the  $F_{COS}$  and  $F_{COS}$  are the  $F_{COS}$  are the  $F_{COS}$  are the  $F_{COS}$  and  $F_{COS}$  are the F

**Figure 10.** Predicting  $LRU_{can}$  by linear regression for the Mieming location. (a) illustrates the results for August 2023, (b) for July 2023. The results of the physical model and our linear regression model are shown, together with the prediction from the leaf-scale regression equation used in Lai et al. (2024).  $LRU_{leaf}$  for a sunlit top layer leaf and a shaded bottom layer leaf are also included. The results of weighting  $LRU_{leaf}$  with the shaded and sunlit leaf area index in each canopy layer is also shown, as well as the result of weighting  $LRU_{leaf}$  with the shaded and sunlit vegetation  $CO_2$  uptake fluxes in each layer. Observations of sunlit leaves from branch bag measurements are indicated by the black stars. The error bars are calculated as +/- one standard deviation of the observed  $LRU_{leaf}$  values over the periods we average. No error bar is shown when only one measurement was available. The first 10 timesteps (minutes) of the physical and new linear regression model are not shown, to reduce potential numerical noise.

**Figure A1.** Posterior correlations of the parameters optimised for Hyytiälä, July 2015. Information on the procedure to estimate the correlations can be found in Bosman and Krol (2023). The shown correlations are marginal correlations and not partial correlations.

**Figure A2.** Modelled and observed ecosystem scale relative uptake (ERU) for the August 2023 Mieming optimisation. Note that the modelled ERU shows a strong change in the early morning, ranging from strongly negative numbers to large positive numbers. This can be explained by the behaviour of the CO<sub>2</sub> flux. As the CO<sub>2</sub> flux occurs in the denominator of Eq. 2, a change from a small positive number to a small negative number leads to a large change in ERU. The error bars for ERU are obtained in the same way as for LRU<sub>can</sub>.

Figure A3. As Fig. 10, but now for the Aug 2023 Mieming optimisation that re-optimises photosynthesis parameters and  $\alpha_{giCOS}$ .

**Table A1.** Used observation streams in the July 2015 Hyytiälä optimisation, together with the posterior partial reduced chi-square statistic of each observation stream (Bosman and Krol, 2023). Note that e.g. the sensible heat flux is measured at approximately 24 m height. In the model the flux at this height is not calculated, the model output we compare with these measurements with is the flux between the top canopy layer and the overlying mixed layer.

| Name                | Description                                    | Units in model                                            | Partial reduced chi-square statistic |
|---------------------|------------------------------------------------|-----------------------------------------------------------|--------------------------------------|
| CO2 <sub>16.8</sub> | CO <sub>2</sub> mole fraction at 16.8 m height | ppm                                                       | 0.481                                |
| $CO2_{8.4}$         | CO <sub>2</sub> mole fraction at 8.4 m height  | ppm                                                       | 0.538                                |
| CO2 <sub>4.2</sub>  | CO <sub>2</sub> mole fraction at 4.2 m height  | ppm                                                       | 0.429                                |
| $COS_{14}$          | COS mole fraction at 14 m height               | $\operatorname{ppb}$                                      | 0.175                                |
| $COS_4$             | COS mole fraction at 4 m height                | $\operatorname{ppb}$                                      | 0.278                                |
| $COS_{0.5}$         | COS mole fraction at 0.5 m height              | $\operatorname{ppb}$                                      | 0.901                                |
| $q_{16.8}$          | Specific humidity measured at 16.8 m height    | $\mathrm{kg}\mathrm{kg}^{-1}$                             | 0.061                                |
| $q_{8.4}$           | Specific humidity measured at 8.4 m height     | $\mathrm{kg}\mathrm{kg}^{-1}$                             | 0.061                                |
| $q_{4.2}$           | Specific humidity measured at 4.2 m height     | $\mathrm{kg}\mathrm{kg}^{-1}$                             | 2.697                                |
| $q_{125}$           | Specific humidity measured at 125 m height     | $\mathrm{kg}\mathrm{kg}^{-1}$                             | 0.277                                |
| $q_{50.4}$          | Specific humidity measured at 50.4 m height    | $\mathrm{kg}\mathrm{kg}^{-1}$                             | 0.284                                |
| $CO2_{125}$         | CO <sub>2</sub> mole fraction at 125 m height  | ppm                                                       | 0.132                                |
| $CO2_{50.4}$        | CO <sub>2</sub> mole fraction at 50.4 m height | ppm                                                       | 0.152                                |
| $COS_{125}$         | COS mole fraction at 125 m height              | $\operatorname{ppb}$                                      | 0.138                                |
| $COS_{23}$          | COS mole fraction at 23 m height               | $\operatorname{ppb}$                                      | 0.282                                |
| $T_{67}$            | Temperature at 67.2 m height                   | K                                                         | 0.004                                |
| $T_{16.8}$          | Temperature at 16.8 m height                   | K                                                         | 0.130                                |
| $T_{4.2}$           | Temperature at 4.2 m height                    | K                                                         | 0.233                                |
| H                   | Sensible heat flux at $\approx 24 \text{ m}$   | ${ m Wm^{-2}}$                                            | 1.126                                |
| LE                  | Latent heat flux at $\approx 24 \text{ m}$     | ${ m Wm^{-2}}$                                            | 0.637                                |
| $F_{\rm CO2}$       | $CO_2$ flux at $\approx 24$ m height           | ${\rm mgCO_2m^{-2}s^{-1}}$                                | 0.129                                |
| $F_{\text{COS}}$    | COS flux at $\approx 23$ m height              | $\rm ppbms^{-1}$                                          | 0.194                                |
| $F_{\rm COS, soil}$ | Soil COS flux                                  | $\rm molCOSm^{-2}s^{-1}$                                  | 0.049                                |
| $F_{\rm CO2, soil}$ | Soil respiration                               | $\mathrm{mol}\mathrm{CO}_2\mathrm{m}^{-2}\mathrm{s}^{-1}$ | 0.086                                |
| $U_{16.8}$          | Horizontal wind speed at 16.8 m height         | ${ m ms}^{-1}$                                            | 0.588                                |
| $U_{8.4}$           | Horizontal wind speed at 8.4 m height          | ${ m ms^{-1}}$                                            | 0.017                                |