# Peer review of "Relative uptake of carbonyl sulphide to CO2: insights from a coupled boundary layer - canopy inverse modelling framework"

_EGUsphere, 2025_

## Referee Comment (RC2)

**Reviewer Comments**

**Summary**

The authors present their work developing a coupled modelling framework simulating the exchange of $CO_2$, OCS and $H_2O$ between surface processes and the lower atmosphere – including throughout the canopy. Parameterisations in the coupled model are adjustable and are done so by assimilating observations using an inverse modelling framework. The aim of this work is to improve representation of the ratio of exchange between $CO_2$ and OCS, the leaf relative uptake (LRU), for needleleaf boreal forest biomes. The methodology used here is robust and is exceptionally important in the field of OCS and the carbon cycle in general. The comparison across multiple sites and variations within-canopy are of particular note and interest for the OCS and carbon cycle community.

**Major Comments**

1. The complexity of the canopy model, SilCan, used in the modelling of PAR at the top and throughout the canopy, is somewhat downplayed. An extensive summary is provided in the supplement, but little is discussed of its evaluation. While the results of this work do somewhat suggest it is performing adequately, i.e. LRU values are sensible in the canopy. A more elaborate evaluation of its performance is recommended or should be presented if done so. How does it compare with other canopy models? Has there been any comparison with estimates of PAR from remote sensing or in-situ observation? Has a separate publication specifically detailing this model been considered?

2. To what extent does the absence of advection and chemistry modelling impact the applicability of the resulting LRU even for the same biome elsewhere? It is mentioned that advection is set to zero for all simulations. A sentence or two is required to highlight the limitations associated with this. For example, on the day scale, changes to air temperature or precipitation would substantially affect LRU. How does this scale up to application of LRU on an annual basis?

3. The COS mole fraction and COS flux in Figure 2 (b, d and f) are exceptionally noisy compared to the other variables (additionally, I don't understand an increase in mole fraction around midday). While this is highlighted in the text, has the full extent of this knock-on effect been considered? This is likely contributing to the positive bias seen in the LRU output. But to what extent does this allow for other variables to dominate in the inversion calculations?

4. Section 3.1.3 contains a particularly interesting discussion and explanation of how the modelled variables affect one another. However, it's very difficult to follow, particularly as the plots are not particularly easy to interpret. Some specific points on this include:

a. In Figure 3: having a solid blue line in multiple panels on the same plot would often suggest they are the same variable, but for different circumstances, say for morning, noon and evening or similar, but in this case, they are for different variables all together, this could be misleading or confusing.

b. Line 305: "This can to a large extent be explained by the profiles of absorbed PAR (Fig. 3b, **red and green dashed lines**).". Include specific references to the plot, so it is easier for the reader to identify.

I recommend a restructuring or re-write of this section. And making the colours more reader friendly and interpretable (specifically with regards to colour-blindness). Or even adjust the structure of the plots. A final note is to consider moving the legends off the plot entirely.

5. The parameterisations for estimating LRU from Kooijmans et al. (2019) are used to estimate LRU – Lai24. As these results seem to perform better at Hyytiälä, which is addressed around Line 510. A more thorough explanation of why this is the case, perhaps with further analysis would be interesting. The shape of the Lai24 model run appears to fit the measurements better, particularly the inflections at the start and end of the day, which is not as apparent in the physical or regression model. Further, the models designed in this work might lead to better correlation values at Mieming, but does this necessarily mean it is performing 'better'? Are there sufficient measurements at Mieming to understand the shape of diurnal LRU? Would we expect the same 'bowl' shaped LRU we see at Hyytiälä here?

6. A few points on discussion and conclusions:

a. More emphasis should be put on the points raised at the end of Section 4.2, such that the inverse modelling framework is suitable to estimate COS and $CO_2$ uptake across different biomes. However, abundant measurements are required at such sites and that the results likely show that a separate exercise of establishing parameterisations at each site is necessary. Until the model achieves improved transferability.

b. A little too much emphasis is put on the physical and regression models outperforming Lai24 at Mieming. Particularly the authors have been slightly overcomplimentary in the performance of the physical and regression models at Mieming site. Figure 10 shows both models and Lai24 still underestimating observations.

c. In conclusions: "*For Hyytiälä, both the physical and regression model generally somewhat overestimated $LRU_{can}$ with respect to the (noisy) observations. We found that the LRU of sunlit top leaves provides a relatively good estimate of $LRU_{can}$, which is encouraging for the use of canopy COS fluxes to estimate canopy $CO_2$ uptake. At the same time, we find that the simple leaf-scale parameterisation obtained in Hyytiälä by Kooijmans et al. (2019), rolled out globally by Lai et al. (2024), does not*

> *perform well in a more southerly needleaf forest (Mieming, Austria)."*. This feels insincere to mention Lai24's underperformance at Mieming, while omitting its good performance at Hyytiälä. Particularly given the third research question raised in Section 1.

The strength of this work is the improved understanding of LRU variations within the canopy and on the development of a simple and well represented regression model. More work is required if the goal is to make model parameterisations transferrable between biomes (in further publications, not this one). While the performance of the physical and regression models at Mieming are slightly overplayed, it is worth noting that results do not necessarily have to be sold as 'good' or 'better' to be interesting and valuable. A little more in-depth discussion on the cause of the discrepancy between the physical model and Lai24, and the performance against measurements would enhance this already well-written publication.

Please see the supplementary document for more in-depth and specific comments. Note they are written up chronologically, not ranked by importance or significance.

**Minor Comments**

1. Title: "Relative uptake of carbonyl sulphide to CO2: insights from a coupled boundary layer - canopy inverse modelling framework". Consider aligning the use of chemical abbreviation and written form.
2. Lines 104-105: overall I think the coupled modelling framework and inverse system could do with a more thorough explanation here. It appears the appropriate publications and documentation have been cited, but an additional sentence or two may help readers outside of the modelling community.
3. Figure 1: The diffuse and direct radiation is a little misleading in this diagram. Assuming the direct radiation is that which is incident directly on the vegetation and diffuse is from reflection and scattering processes in the atmosphere. Having the direct arrows going diagonal through the atmosphere is counter-intuitive to the idea that a direct path would be the shortest path possible. I think resolved if the diffuse radiation arrows emphasise the randomness of scattering processes in the atmosphere. Perhaps coming off the direct beam..?
4. References to the supplementary material should include specific section or equation references, given the length of the supplementary document (examples include 135 and 228)

**Technical Comments**

Line 7: include - includes*

Line 18: optimations – optimisations*

Egusphere-2025-4714
Comments made January 2026

Line 24: Define VPD

Line 38: subscript 2 in CO2 – $CO_2$*

Line: 102-103: add in link to the canopy model in question: "We have added a relatively simple canopy model to the ICLASS framework **(SiLCan, see Section 2.2)**, in order to simulate gases and atmospheric conditions in forest canopies in more detail."

Lines 108-111: as this is directly discussed in Bosman and Krol, 2023, I think it would help the reader to direct them to it specifically, i.e. "see Section 3.1 in Bosman and Krol, 2023".

Line 121: moisture -> $H_2O$. As you are explicitly referring to the tracers, I think it would be best to be clear exactly which molecule you are referring to.

Line 149: Figure -> Fig.

Line 215-216: "The formula for **LRU is found by rearranging Eq. 1**"

Line 237: Define VPD

Line 315: small T for the

Line 409: full stop after Fig - Fig**.**

Check instances of capitalised LAI24, for example in the legend of Figure 10. Important to differentiate between Leaf Area Index (LAI) and Lai24 the parameterisation.